# Molecular Features and Clinical Management of Hereditary Gynecological Cancers

**DOI:** 10.3390/ijms21249504

**Published:** 2020-12-14

**Authors:** Arisa Ueki, Akira Hirasawa

**Affiliations:** 1Center for Medical Genetics, Keio Cancer Center, Keio University School of Medicine, 35 Shinanomachi, Shinjuku-ku, Tokyo 160-8582, Japan; ar7aksm@keio.jp; 2Department of Clinical Genomic Medicine, Graduate School of Medicine, Dentistry and Pharmaceutical Sciences, Okayama University, 2-5-1 Shikata-cho, Kita-ku, Okayama 700-8558, Japan

**Keywords:** hereditary gynecological cancer, multi-gene testing, tumor profiling, hereditary breast and ovarian cancer (HBOC), Lynch, Li–Fraumeni, Cowden, Peutz–Jeghers, syndrome

## Abstract

Hereditary gynecological cancers are caused by several inherited genes. Tumors that arise in the female reproductive system, such as ovaries and the uterus, overlap with hereditary cancers. Several hereditary cancer-related genes are important because they might lead to therapeutic targets. Treatment of hereditary cancers should be updated in line with the advent of various new methods of evaluation. Next-generation sequencing has led to rapid, economical genetic analyses that have prompted a concomitant and significant paradigm shift with respect to hereditary cancers. Molecular tumor profiling is an epochal method for determining therapeutic targets. Clinical treatment strategies are now being designed based on biomarkers based on tumor profiling. Furthermore, the National Comprehensive Cancer Network (NCCN) guidelines significantly changed the genetic testing process in 2020 to initially consider multi-gene panel (MGP) evaluation. Here, we reviewed the molecular features and clinical management of hereditary gynecological malignancies, such as hereditary breast and ovarian cancer (HBOC), and Lynch, Li–Fraumeni, Cowden, and Peutz–Jeghers syndromes. We also reviewed cancer-susceptible genes revealed by MGP tests.

## 1. Introduction

Cancers that accumulate in families have been regarded as familial cancers. However, recent advancements in medical research have led to the re-definition of some familial cancers that are closely associated with genetic factors as hereditary cancers. Many of these arise due to pathogenic germline variants of the causative genes. The two-hit theory was presented in 1971 as a carcinogenic mechanism of autosomal-dominant inherited retinoblastoma [1]. This theory states that a loss-of-function mutation in one copy of a tumor-suppressive, predisposing gene in the germline (first hit), is followed by a somatic mutation (second hit) in another copy of the gene. 

The typical clinical features of hereditary cancers include intrafamily accumulation of specific cancers, juvenile onset, and simultaneous/metachronous multiple cancers such as those with bilateral onset. Most hereditary cancers have autosomal dominant inheritance, with a 50% probability that the pathogenic variant will be passed down to the next generation, regardless of gender. When an individual has a specific genotype, the probability that the trait will be expressed in the body is called penetrance, and this depends on the causative gene. The cumulative risk of hereditary cancers is rarely 100%, and cancers notably do not develop in all individuals harboring the pathogenic variant.

Medical geneticists and genetic counselors can provide support for patients and families as needed if a hereditary cancer is suspected [2,3,4,5]. After explaining the advantages and disadvantages of genetic diagnosis, and obtaining written, informed consent, patients can undergo genetic tests. When causative pathogenic are confirmed by the results of such tests, patients are considered as carriers of pathogenic variants. However, not all genetic tests result in a diagnosis, which might be due to methodological limitations of tests, the involvement of other causative genes, unknown genes, environmental factors, or the patient is negative for a pathological variant. However, even if genetic tests do not reveal pathogenic variants of a gene, the possibility of a hereditary cancer cannot be ruled out, and individuals should be evaluated considering their medical and family history [6].

Precision medicine has recently been advocated, and personalized treatment strategies based on tumor profiling have attracted attention. Vertical cancer treatment of a specific organ has become possible across organs in precision medicine. Molecular tumor profiling is an epochal means of identifying therapeutic targets, and the design of clinical treatment strategies according to biomarkers defined by tumor profiling is becoming a major trend. This review summarizes the characteristics of various genetic tests, current knowledge of gynecological hereditary cancers, and their characteristics and clinical management.

## 2. Hereditary Gynecological Cancers

Cancer can develop in multiple organs and the cause can be a wide variety of hereditary cancer-related genes. Cancers that develop in the ovaries and uterus often overlap with hereditary cancers. Gynecologists play a crucial role in the diagnosis, treatment, and subsequent management of hereditary cancers. This section outlines the typical gynecological hereditary cancers shown in Table 1.

### 2.1. Hereditary Breast and Ovarian Cancer Syndrome

Hereditary breast and ovarian cancer syndrome (HBOC) is diagnosed when a pathogenic variant is identified in the *BRCA1* or *BRCA2* (*BRCA1/2*) genes, which are involved in DNA damage repair. This syndrome tends to develop in younger persons, and cancers of the breast, ovaries, fallopian tubes, and peritoneum typically occur within families [7,8,9]. The syndrome includes high-grade serous ovarian, male breast, and bilateral breast cancers.

Kuchenbaecker et al. reported that the cumulative risk of developing breast and ovarian cancers by the age 80 years is 72% and 44% and 69% and 17% for carriers of the *BRCA1* and *BRCA2* pathogenic variants, respectively [7]. Notably, 10–15% of all ovarian cancers are associated with *BRCA1/2* pathogenic variants. Hirasawa et al. reported that 8.3% and 3.5% of all patients with ovarian cancer in Japan had *BRCA1* and *BRCA2* pathogenic variants, respectively [8]. This indicated that patients with HBOC are being treated for sporadic ovarian cancer. Clinically assessing genetic risk of ovarian cancer is important to ensure the choice of appropriate treatment. Ovarian cancer in the context of HBOC is characterized by a high proportion of serous carcinoma and patients with advanced stage III or higher [9,10,11,12].

Carriers of the *BRCA1/2* pathogenic variant should be managed by risk-reducing surgery, and by surveillance screening [6] for hereditary cancers at an early stage. Guidelines in various countries recommend salpingo-oophorectomy (RRSO) which reduce risk overall [6,13,14,15,16], and of developing ovarian and fallopian tube cancer, and improves prognosis. Rebbeck et al. reported that RRSO for *BRCA1/2* pathogenic variant carriers reduces the risk of developing ovarian and fallopian tube cancer by 79% (HR, 0.21, 95% CI, 0.12–0.39) [17]. However, gynecological surveillance is required because the risk of developing peritoneal cancer persists even after RRSO. Harmsen et al. reported a 3.5% incidence of peritoneal cancer 10 years after RRSO for *BRCA1/2* pathogenic variant carriers [18]. If RRSO is not selected, gynecological surveillance by transvaginal ultrasonography and serum tumor marker CA125 are options; but these have not yet been validated [19].

Poly ADP ribose polymerase (PARP) inhibitors comprise a promising therapeutic strategy for HBOC-related cancers [20]. Both BRCA1/2 and PARP1 are involved in DNA damage repair. If the *BRCA1/2* genes are dysfunctional, DNA repair then depends on PARP1. A PARP inhibitor inhibits the action of PARP1 in HBOC-related cancers in which the *BRCA1/2* gene is dysfunctional, and specifically leads cancer cells to apoptosis. This mechanism (synthetic lethality) is gaining attention. With the introduction of PARP inhibitors for treating breast and ovarian cancers, the presence or absence of the germline *BRCA1/2* pathogenic variant is being determined for appropriate drug selection [21,22,23,24,25,26,27,28,29,30,31,32,33,34]. The BRCA1 and BRCA2 proteins repair DNA double-strand breaks via the homologous recombination repair (HRR) pathway. A homologous recombination deficiency (HRD) is a target for PARP inhibitors, and HRD status now serves as a biomarker for indicating the appropriate time to apply these agents [25,35,36,37]. Various guidelines recommend *BRCA1/2* genetic tests of all ovarian cancers [6,16,38]. Appropriate genetic care should be available to unaffected relatives of a family member who tested positive for *BRCA1/2*.

### 2.2. Lynch Syndrome

Lynch syndrome is a hereditary cancer syndrome caused by germline pathogenic variants in DNA mismatch repair genes (MMR) such as *MLH1, MSH2, MSH6, PMS2,* and *EPCAM*. Families with Lynch syndrome have a high lifetime risk of developing colorectal, endometrial, ovarian, small intestine, ureteral, and renal pelvis cancer, and tend to develop cancer at a young age. The risk of developing cancer in Lynch syndrome differs depending on the causative gene [39].

Lynch syndrome accounts for ~3% of all colorectal cancers and is one of the most common hereditary cancers [40]. The lifetime risk of developing endometrial cancer is comparable to that of colorectal cancer in women with Lynch syndrome. The average age of onset of endometrial cancer in women with Lynch syndrome is 47–55 years, which is younger than in the general population [41]. Therefore, endometrial cancer in a woman with Lynch syndrome becomes a “sentinel cancer,” which is the first diagnosed cancer in that individual [42]. After treatment for endometrial cancer, measures against other cancers such as colorectal cancer might be needed and members of a family in which one person has Lynch syndrome, should also be appropriately surveilled. Although surveillance for endometrial cancer in Lynch syndrome is not supported by evidence, the diagnostic utility of endometrial histology is high, and implementation every 1–2 years is considered [38,43]. In addition, endometrial cancer develops at a younger age in patients with Lynch syndrome, and the prognosis is good. The cumulative lifetime incidence of ovarian cancer in women with Lynch syndrome is 8–20% [44,45], but few reports have described ovarian cancer related to Lynch syndrome. Watson et al. characterized ovarian cancer in Lynch syndrome as follows: prevalent in various histological types, early stage (61% in stage I), average age at diagnosis is 43 years, and comorbid with endometrial cancer in 22% of patients [46,47].

Cancer cells with impaired function caused by two hits on MMR genes characteristically have abnormal replication of repetitive sequences, namely microsatellite instability (MSI) [48]. Tumors with MSI in two or more microsatellite regions are MSI-high (MSI-H), with only one MSI-low (MSI-L) region, and tumors without MSI are classified as microsatellite stable (MSS). Lynch syndrome has been identified in 16.3% of patients with MSI-H tumors [48]. That study also found that most patients with Lynch syndrome had MSI-H/I, and that 36% had MSS tumors. Indeed, among these patients with Lynch syndrome, 71.2% and 78.4% of germline pathogenic variants were detected in *MLH1*, *MSH2*, or *EPCAM* genes in MSI-H/I tumors, but in the lower-penetrance *PMS2* or *MSH6* genes in MSS tumors. Not only the risk of developing cancer, but also the frequency of MSI-H notably differs among MMR genes in Lynch syndrome. If Lynch syndrome is suspected, primary screening should determine whether it meets the Amsterdam II [49], or revised Bethesda criteria [50]. If these criteria are met, secondary screening for MSI or immunohistochemical tests should proceed to confirm MSI-H or the loss of protein expression by MMR genes [38]. Lynch syndrome is diagnosed when subsequent genetic testing reveals a pathogenic germline variant in MMR genes. Many genes involved in carcinogenesis contain microsatellite regions, and the accumulation of abnormalities in these regions results in MSI-H. Immune checkpoint inhibitors (ICI) are particularly effective against tumors with MSI-H and should be effective in Lynch syndrome [51,52].

### 2.3. Li–Fraumeni Syndrome

Li–Fraumeni syndrome is an autosomal-dominant hereditary cancer of juvenile onset caused by the *TP53* gene [53]. Soft tissue sarcoma, osteosarcoma, adrenocortical carcinoma, brain tumors, as well as premenopausal breast, colorectal, and gastric cancer can develop due to Li–Fraumeni syndrome, and the lifetime risk for all cancers almost 100% [54,55,56,57]. Among them, soft tissue sarcoma, brain tumors, and adrenocortical carcinoma develop in childhood; therefore, intervention for family members is important. According to a French study, 14% of patients harbor de novo *TP53* mutations [58]. Even if they do not have a familial history of cancer, Li–Fraumeni syndrome should be considered in specific pediatric patients with adrenocortical carcinomas or choroid plexus tumors, women with breast cancers before reaching the age of 31 years, or multiple primary cancers within the Li–Fraumeni spectrum.

Breast cancer in Li–Fraumeni syndrome affects 54% of the women by the age of 70 years [54,57]. Among women with breast cancer aged < 30 years without a family history of breast cancer, 3–8% have a *TP53* pathogenic germline variant [58,59,60,61], suggesting that Li–Fraumeni syndrome is not uncommon. The average age of onset of breast cancer in Li–Fraumeni syndrome is the early 30s, and surveillance using breast MRI with contrast is recommended from the age of 20 years [6]. In addition, exposure to radiation including therapy might cause secondary cancer in Li–Fraumeni syndrome [6,53]. Total mastectomy is recommended to avoid radiation therapy for breast cancer patients with Li–Fraumeni syndrome [62], and should be considered when deciding surgical procedures. The NCCN guidelines for Li–Fraumeni syndrome describe that the *TP53* pathogenic variant does not increase the risk of ovarian cancer [6], but ovarian cancer has developed in patients with Li–Fraumeni syndrome [57,63].

The *TP53* gene is referred to as, “the guardian of the genome,” and its function is lost in various cancers. Although *TP53* is one of the most frequently encountered genes in tumor profiles, it rarely has pathogenic variants in the germline [64]. Therefore, it is difficult to diagnose only by tumor profiling, and individual measures should be taken according to the medical and family history of each patient. Even if tumor profiling reveals *TP53*, no treatment is yet available. However, clinical trials of TP53-targeting compounds are underway [65].

### 2.4. Cowden Syndrome

Cowden syndrome is a multiple hamartoma syndrome caused by a germline pathogenic variant of *PTEN*, with a 77–85% lifetime risk of breast cancer in women [66]. The major criteria for clinical diagnosis include breast cancer, epithelial (especially follicular) thyroid cancer, gastrointestinal hamartoma, macrocephaly, and endometrial cancer [6,67]. Risk of endometrial cancer in Cowden syndrome is increased within the ages of 30 and 49 years, and the lifetime risk is 28% [68]. In addition, benign uterine fibroids often occur, but they are not included in the clinical diagnostic criteria due to insufficient evidence [6]. The utility of endometrial cancer screening as gynecological surveillance for Cowden syndrome has not been established. However, although the value of surveillance is limited, endometrial cytology/histology and transvaginal ultrasonography every 1–2 years should be considered for women with Cowden syndrome after the age of 30–35 years [6]. 

Somatic pathogenic variants in the *PTEN* gene have been detected in various tumor profiles [69]. Since *PTEN* is involved in the PI3K/Akt/mTOR pathway, mTOR inhibitors should be effective. However, the effects of such inhibitors have not yet been fully elucidated [70,71,72,73]. 

### 2.5. Peutz–Jeghers Syndrome (PJS)

The autosomal dominant genetic disorder, PJS, is characterized by multiple hamartoma polyps in the gastrointestinal tract and pigmentation of the skin mucosa [74], and *STK11* is the causative gene [75,76]. Hamartomatous polyps of PJS are found in the stomach, small intestine, large intestine, and elsewhere, and melena and intestinal obstruction can occur. Epithelial, malignant colorectal, gastric, pancreatic, breast and other types of cancer are related to PJS [77]. Pigmentation is often found on the fingers, lips, and oral cavity of children; hence, a clinical diagnosis is relatively straightforward. However, pigmentation disappears with age [77].

Gynecological ovarian tumors are PJS-related. Ovarian tumors with PJS are mainly of the sex-cord with annular tubules (SCAT), but they can also be associated with other ovarian malignancies. The risk of SCAT in PJS is 21%, and the average age at diagnosis is 28 years [78], which cannot be overlooked. Minimal deviation adenocarcinoma (MDA) and lobular endocervical glandular hyperplasia (LEGH) in PJS are referred to as PJS-related cervical tumors [79]. Both MDA and LEGH are malignant tumors of the cervix formerly known as adenoma malignum, and they have a poor prognosis. The 5-year survival rate of MDA is 42% [80]. The risk of MDA morbidity in the general population is < 1%, but the lifetime risk in patients with PJS is 10%, and the average age of onset is 34–40 years [77,78,81]. The fact that MDA can be difficult to diagnose should be considered in the surveillance of female patients with PJS. Since PJS is often diagnosed in childhood, surveillance should be continued in collaboration with various clinical departments. 

Abnormalities of the *STK11* gene in cancers have recently been associated with resistance to ICI [82,83].

### 2.6. Other Cancer-Susceptible Genes

Genes with high and moderate susceptibility for hereditary cancers have recently been identified and classified. Genes with low susceptibility are also classified as causes of multifactorial carcinogenesis. The risk of developing cancer is high in patients with highly and moderately susceptible genes [11,84,85], and many genes in the NCCN guidelines can be referred to for management [6,43]. Although the penetration rate is low compared with HBOC and Lynch syndrome, hereditary cancers caused by other genes associated with cancer susceptibility cannot be overlooked. For example, the *BRIP1, PALB2, RAD51C, RAD51D*, and *BARD1*, as well as the *BRCA1/2*, *MLH1, MSH2, MSH6,* and *PMS2* genes are associated with hereditary ovarian cancer [11]. These genes are also thought to be involved in HRD, and PARP inhibitors should be effective in patients with these cancer-susceptible genes [28,35,37,86,87]. 

*DICER1* causes a cancer predisposition syndrome and *DICER1* syndrome is an autosomal dominant genetic disorder characterized by pleuropulmonary blastoma (PPB), multinodular goiter, cystic nephroma, Sertoli–Leydig cell tumors of the ovary (SLCT), and other rare tumors. Pleuropulmonary blastoma is the most prevalent primary lung malignancy in children, and those at highest risk for clinically significant PPB are aged < 7 years [88]. Ovarian tumors associated with *DICER1* diagnosed at a median age of 16.9 years, sometimes develop synchronous or metachronous contralateral tumors [88]. Stewart et al. reported that 24 (21.2%) of 113 female carriers of *DICER1* developed SLCT [89]. The *DICER1* gene encodes DICER protein, which is an RNase III enzyme that functions in micro-RNA (miRNA) processing [90]. Targeted therapy against *DICER1* is not yet available. However, a few reports have described these rare cancer-predisposing genes, and the latest information on clinical management should be reviewed, because caution is required when dealing with moderate- and low-grade susceptibility genes. The numbers of rare hereditary cancer-related genes judged as variants of uncertain significance (VUS) is likely to increase [84,91,92], and additional confirmation might be required for interpretation. 

## 3. Genetic Testing for Hereditary Cancers

Genetic test results targeting hereditary tumor-related genes are usually analyzed using next-generation sequencing. However, further investigation is occasionally needed to conclude a correct diagnosis. Figure 1 shows that genetic alterations include single nucleotide variants (SNV), insertion and deletion of bases (indels), splicing variants in non-coding intron regions, copy number variants (CNV), and epigenetic changes, such as DNA hypermethylation and histone modification. Here, we outline genetic alterations and analyses.

Single nucleotide variants have a change in one base (Figure 1a). Some synonymous SNV encode the same amino acid and have no significant effects, whereas some missense variants have significant effects due to amino acid substitutions. A variant is pathogenic if changes affect the three-dimensional structure of amino acids. When encoded stop codon and produces a truncated protein with altered amino acids, they are called nonsense variants and are often pathogenic. Some frameshift variants move the reading frames of encoded amino acids by inserting or deleting several bases. 

In addition, if a slightly larger deletion/insertion in the exon or gene (Figure 1b) causes difficulties with detecting changes by next-generation sequencing, Multiplex Ligation-dependent Probe Amplification (MLPA) can detect changes in copy numbers. Gene amplification (Figure 1c) and the loss of heterozygosity (LOH; Figure 1d) also occur in cancers. Analyzing copy numbers is useful for detecting LOH, and gene deletions and duplications can be analyzed using fluorescence in situ hybridization (FISH). Even if a change occurs in a non-coding intron, when variants affect the splice site, exon skipping leads to pathogenic protein formation (Figure 1e). Targeted RNA analysis might be required in the event of exon-skipping.

The hypermethylation of DNA and histone modification are epigenetic alterations (Figure 1f) that control gene expression without changing the DNA sequence. Many genes have a region of CpG islands near the promotor region upstream of a target gene. Hypermethylated DNA is often identified in cancers, and gene expression can be controlled by methylating CPG islands. Because DNA hypermethylation cannot be detected by next-generation sequencing, the methylation status of the promoter region should be analyzed. Histone modification is a complex mechanism in which histone acetylation and methylation respectively opens and closes the chromatin structure to activate and switch off transcription.

Consequently, genetic testing using various means can not only diagnose hereditary tumors, but also facilitate pharmacogenomics and the design of personalized therapy for cancer patients. The number of hereditary cancer-related genes diagnosed by tumor profiling as germline findings is increasing. Here, the characteristics of each genetic test are described.

### 3.1. Genetic Tests for Diagnosing Hereditary Cancers 

Single genes have historically been tested based on the most likely hereditary cancer of a patient. With the widespread advent of next-generation sequencing, MGP has become the mainstream genetic test as is more rapid and cost-effective. Judgments were issued in the USA to invalidate the patent for *BRCA1/2* genetic tests monopolized by Myriad Genetics during 2013 [93,94]. Thereafter, several companies have entered the market and now provide various MGP tests. 

A set of genes that are considered to be related to hereditary cancers can be simultaneously analyzed using MGP. The introduction of MGP tests should increase the numbers of individuals diagnosed with pathogenic variants in genes associated with hereditary cancers that hitherto could not be identified by conventional single-gene tests. In fact, MGP tests replaced *BRCA1/2*-only tests in 2014 [91]. The spread of MGP tests will reduce the number of misdiagnosed hereditary cancers. 

On the other hand, the number of patients with rare hereditary cancers will increase, even though they were not suspected before genetic tests. Therefore, MGP tests should be applied with reference to the most recent NCCN guidelines for the management of rare hereditary cancer-related genes. The revised NCCN guidelines (2020) [43,95], caused a major paradigm shift as the description changed to consider MGP tests first among genetic tests.

According to the guidelines of the American College of Medical Genetics and Genomics (ACMG) [96], the results of genetic tests are classified as: pathogenic, likely pathogenic, benign, likely benign, and variant of uncertain significance (VUS). The VUS classification means that pathogenicity cannot be determined, despite some variants in the gene. Genetic management based on VUS results is not recommended.

Multi-gene panels target multiple hereditary cancer-related genes. As the number of MGP tests increase, the number of rare hereditary cancer-related genes judged as VUS will also probably increase, and additional confirmation might be required for appropriate interpretation [84,85,97,98]. Furthermore, whether targeted genes can be analyzed by MGP should be confirmed because individual manufacturers will have different offerings, and not all targeted genes are clinically compatible. Thus, MGP tests must be accompanied by pre- and post-test genetic counseling based on genetic expertise [6,43].

### 3.2. Genetic Tests for Pharmacogenetics and Personalized Therapy

Several genetic tests can be conducted to decide appropriate treatment for patients. In December 2014, the PARP inhibitor, Olaparib, was approved by the European Medicines Agency (EMA) and the US Food and Drug Administration (FDA), after another drug for *BRCA1/2* mutation-positive ovarian cancer [23]. Accordingly, individuals with germline pathogenic variants in the *BRCA1/2* gene are indicated for PARP inhibitors, and HBOC can be simultaneously diagnosed, so it is essential to provide appropriate information to patients and family members.

Indications for ICI, regardless of the type of cancer, have been determined by tests of MSI, and ICI are indicated treating for MSI-H solid tumors [52,99]. However, Lynch syndrome is also a possibility in patients with MSI-H [48]. Thus, careful assessment based on family and medical history is required. 

The features of a *BRCA1/2* functional deletion have been called BRCAness, the definition of which is ambiguous. Thus many tests have been proposed, such as the HRD score [100], the COSMIC mutational signature #3 [101], and the LOH status of the *BRCA1* or *BRCA2* locus [102]. The PARP inhibitor, Niraparib, is useful for late-line treatment of ovarian cancer, when HRD status serves as a biomarker [103]. Germline pathogenic variants in *BRCA1/2* result in HRD, which can be exploited by PARP inhibitor as a matter of course. Although the HRR pathway involves numerous genes, HRR pathway genes other than *BRCA1/2* such as *PALB2*, *RAD51*, and *ATM,* are candidates that might be effective for PARP inhibitor [36,104,105,106,107]. Furthermore, direct sequencing of HRR pathway genes can predict responsiveness to platinum and PARP [108]. Confirmation of HRD might reveal hereditary tumors. 

Thus, the important issue is the appropriate approach to the possibility of hereditary cancers revealed by treatment indications. Collaboration with genetic experts is important, as is linking appropriate genetic counseling and hereditary cancer management when conducting genetic tests [43].

### 3.3. Tumor Profiling for Precision Medicine

The purpose of tumor and gene profiling in individual cancer tissues is to control cancers with personalized therapeutic strategies targeting driver genes [109,110]. On the other hand, the possibility of hereditary cancer-related genes can be clarified by comparison with normal sites as controls during tumor profiling. The amount of incidental germline findings discovered through tumor profiling is increasing [111,112]; a 5–15% chance of germline findings is associated with hereditary cancers by tumor profiling [113,114].

The ACMG (2020) issued a statement on presumed germline pathogenic variants (PGPV) that can be revealed from tumor tests [111]. With respect to the importance of germline findings in tissues, they state that, “Identifying germline pathogenic variants can inform future cancer risks, cancer surveillance, and prevention options for the patient and family members. In addition, germline genetic information, independent of somatic variation, can influence the choice of targeted therapy for a tumor.” Germline pathogenic variants in *BRCA1/2* are informative as they confirm eligibility for treatment with PARP inhibitors.

Germline findings identified by tumor tests indicate that cancers are caused by genetic germline variants that might be shared by the families of patients. The European Society of Medical Oncology (ESMO) Precision Medicine Working Group (PMWG) recommended germline-focused analysis of tumor-only sequences in 2019 [64]. Thresholds of 20% and 30% VAF for small insertions/deletions and SNV, respectively, and limiting target genes for germline-focused tumor analysis to 27 (*BRCA1, BRCA2, BRIP1, MLH1, MSH2, MSH6, PALB2, PMS2, VHL, RAD51C, RAD51D, RET, SDHA, SDHAF2, SDHB, SDHC, SDHD, TSC2, MUTYH, RB1, APC, FLCN, FH, BAP1, POLE, TP53,* and *NF1*), were proposed to narrow variants requiring follow-up germline tests.

By disclosing these germline findings, useful information can be provided to patients and their families, appropriate surveillance methods for early detection and early treatment for cancers can be suggested, indications for risk-reducing surgery can be discussed, and appropriate treatment for individual patients can be selected [114]. When germline alterations are suggested, genetic counselors can organize information considering treatment strategies and their impact on family members. Co-operation with genetic experts is necessary for additional confirmation.

## 4. Conclusions

The introduction of multigene panels has enabled the simultaneous analysis of multiple genes. In addition, a wider range of analyses using tumor profiling to target germline and somatic variants, has facilitated fewer omissions. Whole exome and genome sequencing will become routine in the near future, as analytical technology rapidly advances. However, consensus about target genes for MGP tests and tumor profiling has not been reached, so the situation established by each test institution or facility should be determined. Recognizing common target genes will become essential, and if somatic variants are revealed by tumor profiling, appropriate treatment methods should be designed.

A definitive diagnosis of the causative gene of a hereditary cancer is characterized as being lifelong, affecting the family, and predictive of cancer onset. In addition, patients with hereditary cancer must be carefully managed. However, genetic information could be useful for subsequent treatment selection and actionable information about diseases. When a germline pathogenic variant associated with a hereditary cancer-related gene is detected by tumor profiling, treatment should be individualized for each patient. Clinicians might be reluctant to definitively diagnose hereditary cancers, given the effects on family members. However, medical intervention will be appropriate if clinicians and patients both recognize that hereditary cancer-related genes are important to know as they will lead to primary cancer prevention for family members. A thorough understanding of hereditary cancers should allow clinicians to use tumor profiling information as a useful tool and provide their patients with optimal medical care.

## Figures and Tables

**Figure 1 ijms-21-09504-f001:**
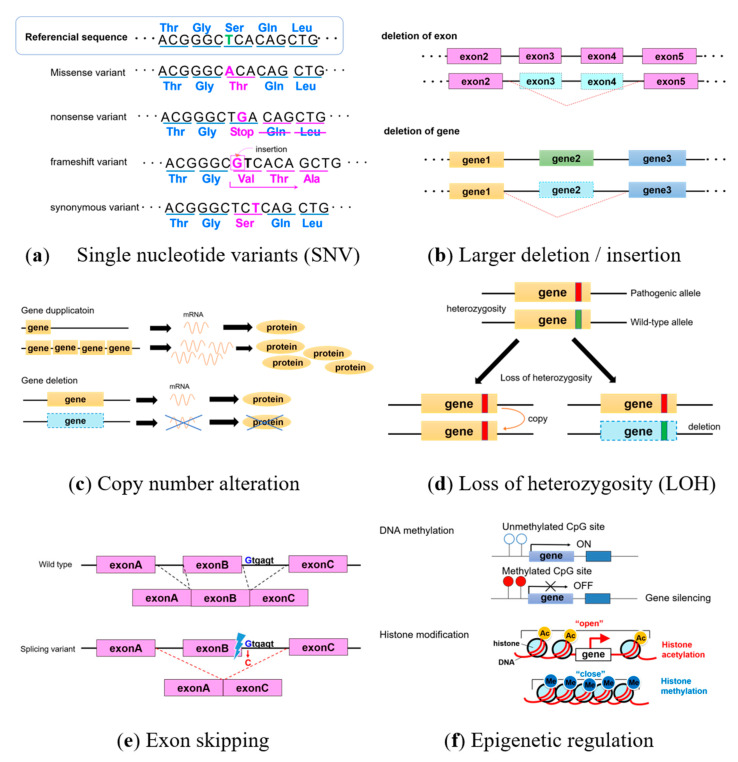
Genetic alterations. (**a**) Single nucleotide variants, (**b**) large deletion/insertion in exon or gene, (**c**) copy number alteration, (**d**) loss of heterozygosity, (**e**) exon skipping, and (**f**) epigenetic regulation.

**Table 1 ijms-21-09504-t001:** Molecular feature and clinical management of hereditary gynecological cancers.

Hereditary Gynecological Cancer	Causative Genes	Related Gynecological Cancers	Related Non-Gynecological Cancers	Drug Sensitivity
HBOC	*BRCA1, BRCA2*	Ovarian cancer	Breast cancer Prostate cancer Pancreatic cancer	PARP inhibitors
Lynch syndrome	*MLH1, MSH2, MSH6, PMS2, EPCAM*	Endometrial cancer Ovarian cancer	Colorectal cancer Small intestine cancer Ureteral cancer Renal pelvis cancer	ICI
Li–Fraumeni syndrome	*TP53*	Insufficient evidence of ovarian cancer	Soft tissue sarcoma Osteosarcoma Premenopausal breast cancer Colorectal cancer Gastric cancer Adrenocortical carcinoma Brain tumor	-
Cowden syndrome	*PTEN*	Endometrial cancer	Breast cancer Epithelial thyroid cancer ^1^ Gastrointestinal hamartoma	-
Peutz–Jeghers syndrome	*STK11*	Ovarian tumor (SCAT) Cervical tumor (MDA, LEGH)	Colorectal cancer Gastric cancer Pancreatic cancer Breast cancer	-
Other cancer-susceptible genes	*RAD51C,* *RAD51D* *Etc.*	Ovarian cancer	Various cancers	HRD status→PARP inhibitors
	*DICER1*	Sertoli–Leydig cell tumor (SLCT) of the ovary	Pleuropulmonary blastoma	-

^1^ Particularly follicular thyroid cancer. Abbreviations: HBOC, hereditary breast and ovarian cancer syndrome; HRD, homologous recombination deficiency; ICI, immune checkpoint inhibitor; LEGH, lobular endocervical glandular hyperplasia; MDA, minimal deviation adenocarcinoma; PARP, poly ADP ribose polymerase; SCAT, sex-cord tumor with annular tubules; SLCT, Sertoli–Leydig cell tumor.

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
