# Peer review of "Molecular Features and Clinical Management of Hereditary Gynecological Cancers"

_ijms, 2020, doi:10.3390/ijms21249504_

Round 1

Reviewer 1 Report

This manuscript reviews an important issue in hereditary gynecological cancers. Although the topic addressed is a well-written, interesting, and useful contribution, I think that there are several improvements that should be made before publication in International Journal of Molecular Sciences.

  1. 2.1. Hereditary breast and ovarian cancer syndrome: HBOC (line 14). “serous adenocarcinoma” should be changed to “serous carcinoma”
  2. 2.1. Hereditary breast and ovarian cancer syndrome: HBOC (line 34). “BRCA1 and BRCA2” should be changed to “BRC1/2)
  3. 2.2. Lynch syndrome (line 10). “the general risk average” is unclear, and needs to be clarified.
  4. 2.2.Lynch syndrome (line 11–12). “Therefore, ……. in the individual.” I think this sentence does not convey the meaning the authors intended. Please revise this sentence.
  5. Please show, with appropriate data, how accurate an MSI test is in Lynch syndrome section.
  6. 2.3. Li-Fraumeni syndrome. Although Li-Fraumeni syndrome is inherited in an autosomal dominant manner, there are a few cases of de novo TP53 mutations. Please discuss this matter including references.
  7. 3.3. Tumor profiling. Please discuss presumed pathogenic germline variants (PPGVs) with typical genes.

Reviewer 2 Report

I have read with interest the paper by Arisa Ueki and Akira Hirasawa "Molecular Features and Clinical Management of Hereditary Gynecological Cancers". In this manuscript, the authors summarized the current findings about the Hereditary Gynecological Cancers. In my opinion the paper is not scientifically accurate, complete and fully comprehensive to the reader. The manuscript should be reviewed with greater accuracy, possibly paying more attention to the specific molecular features and precision medicine concepts. There are some inaccuracies that should be correct and some arguments that should be implemented. The discussion appears confusing, the authors should better organize the structure of the paper.

The comments below are by way of example, the manuscript must be altered in a manner more understandable and accurate.

Introduction: “conflict allele” is not an exact term that can be applied to the Knudson hypothesis

Introduction: "Penetrance of carcinogenesis" does not appear to be currently used. Penetrance: Describes how likely it is that a person who has a certain disease-causing mutation (change) in a gene will show signs and symptoms of the disease; Carcinogenesis: The process by which normal cells are transformed into cancer cells (NCI Dictionaries).

Introduction: use of the word “client”.  Although the term “client” is not substantially incorrect (Shevell MI. What do we call 'them'?: the 'patient' versus 'client' dichotomy. Dev Med Child Neurol. 2009 Oct;51(10):770-2), it is preferable, in oncology, to use the term patient

Table 1:    in this table and in the following text the authors should also include DICER1 (OMIM*606241; https://www.ncbi.nlm.nih.gov/books/NBK196157/)

paragraph 2. “Hereditary Gynecological Cancers”:  since the title includes "Molecular Features" the authors should dwell on technical notes and peculiarities of the mutational analysis relating to the genes described. For example, rather than dwelling on single-gene and multi-gene-panel (see below) the authors should describe the molecular alterations as large deletions, CNVs or methylation of these genes.

paragraph 2.5:   “LKB1 and STK11 have been identified as the causative genes of PJS”. LKB1 and STK11 are not two genes, they are the same gene (OMIM *602216)

paragraph 3.1-3.2:   The authors use two paragraphs to describe Single-gene testing and Multi-Gene Panel (MGP) testing. It seems difficult to understand the reason for this division since the authors themselves declare that "Nowadays, MGP testing is the mainstream test as genetic testing has become cheaper and faster with the widespread use of next generation sequencing". Some pharmacogenetic topics are included in the paragraph "Single-gene testing". Authors should instead insert a specific paragraph on pharmacogenetics and personalized therapy based on the results obtained from the mutational analysis. For example, the use of PARP inhibitors is also expects in the case of BRCAness, which can investigated only using Multi-Gene Panels (Byrum AK et al Defining and Modulating 'BRCAness'. Trends Cell Biol. 2019 Sep;29(9):740-751; Lin PH et al. Using next-generation sequencing to redefine BRCAness in triple-negative breast cancer. Cancer Sci. 2020 Apr;111(4):1375-1384; Lin PH et al.. Using next-generation sequencing to redefine BRCAness in triple-negative breast cancer. Cancer Sci. 2020 Apr;111(4):1375-1384). 

In addition some linguistic corrections are necessary.

Round 2

Reviewer 2 Report

The authors have satisfactorily modified the text as required by the previous revision and have clarified in a precise manner all the doubts concerning the paper.